# Identifying the Roles of Healthcare Leaders in HIT Implementation: A Scoping Review of the Quantitative and Qualitative Evidence

**DOI:** 10.3390/ijerph17082865

**Published:** 2020-04-21

**Authors:** Elina Laukka, Moona Huhtakangas, Tarja Heponiemi, Outi Kanste

**Affiliations:** 1Social and Health Systems Research Unit, Finnish Institute for Health and Welfare, 00271 Helsinki, Finland; tarja.heponiemi@thl.fi; 2Research Unit of Nursing Science and Health Management, University of Oulu, 90230 Oulu, Finland; moona.huhtakangas@oulu.fi (M.H.); outi.kanste@oulu.fi (O.K.)

**Keywords:** health information technology (HIT), implementation, healthcare, leader, scoping review

## Abstract

Despite major investment, health information technology (HIT) implementation often tends to fail. One of the reasons for HIT implementation failure is poor leadership in healthcare organisations, and thus, more research is needed on leaders’ roles in HIT implementation. The aim of the review was to identify the role of healthcare leaders in HIT implementation. A scoping review with content analysis was conducted using a five-step framework defined by Arksey and O’Malley. Database searches were performed using CINAHL, Business Source Complete, ProQuest, Scopus and Web of Science. The included studies were written either in English or Finnish, published between 2000 and 2019, focused on HIT implementation and contained leadership insight given by various informants. In total, 16 studies were included. The roles of healthcare leaders were identified as supporter, change manager, advocate, project manager, manager, facilitator and champion. Identifying healthcare leaders’ roles in HIT implementation may allow us to take a step closer to successful HIT implementation. Yet, it seems that healthcare leaders cannot fully realise these identified roles and their understanding of HIT needs enforcement. Also, healthcare leaders seem to need more support when actively participating in HIT implementation.

## 1. Introduction

Globally, the importance of health information technology (HIT) has been highlighted in several healthcare programmes [1,2] and its value is still growing due the pressures that challenge healthcare today. A growing number of HITs have great potential for improving the quality, safety, patient-centredness and cost-effectiveness of care [3,4]. However, digitalisation is slower and more complicated in healthcare settings compared to other fields of business [5,6,7]. In addition, despite major investments [7], HIT implementation tends to fail more often in healthcare settings [6,8,9]. Moreover, cost savings and improvements in the quality of care through HIT implementation may not always be obvious. One of the reasons for HIT implementation failure is poor leadership in healthcare organisations [3,8,10]. 

Identifying implementation determinants such as barriers or facilitators is important, since recognising them may positively improve implementation outcomes [11]. Implementation facilitators and barriers are a consistent focus in the research literature, and findings suggest that healthcare leadership is one of the important implementation determinants [6,12]. For example, lack of support from leaders has been recognised as one of the major barriers in implementation [3,13,14]. Ingebrigsten et al. [15], who conducted a systematic literature review on the impact of clinical leadership on HIT adoption, identified seven leadership behaviours that were associated with successful outcomes in HIT adoption: (1) communicating clearly about visions and goals, (2) providing support, (3) establishing a governance structure, (4) establishing training, (5) identifying and appointing champions, (6) addressing work process change, and (7) follow up. In the current review, healthcare leadership is widely understood, and it includes both leading people and managing systems and structures [16].

It is well-known that HIT implementation requires strong leadership [17,18], yet leaders are not always aware of their roles in HIT implementation [18]. Due to the changing environment and advances in healthcare technology, the roles of healthcare leaders have expanded [19]. Traditionally, it has been part of the healthcare leader’s role to have competency in clinical health services and management [19], but now, leaders must also possess knowledge of technologies related to health information [20,21]. Because leaders can have a positive or negative impact on HIT implementation [6,12,15], more research is needed concerning the role of leaders in HIT implementation [18,22,23], taking all leadership levels into account [22,24,25]. 

In the current review, HIT is understood as a technology “used within a healthcare organisation to facilitate communication, integrate information, document healthcare interventions, perform record keeping, or otherwise support the functions of the organisation” [26]. HIT implementation, in turn, is understood as a wide-ranging process that includes planning of the service and implementation; HIT adoption by healthcare consumers and professionals; and establishment of the service and monitoring [5]. Thus, implemented HITs may include, for example, electronic medical or health records (EMR/EHR) [27,28] and services for self-treatment and digital value [29]. 

The aim of the review was to identify the role of healthcare leaders in HIT implementation. The following research question was addressed: What roles of healthcare leaders can be identified in HIT implementation? Answering this research question may aid organisations to implement HIT more successfully, provide new insights, and identify research gaps for future HIT implementation studies.

## 2. The Review

### 2.1. Design

The five-step framework defined by Arksey and O’Malley [30] was adopted and content analysis was used. The review was conducted using the following steps: (1) Identifying a research question, (2) identifying relevant studies, (3) study selection, (4) charting the data, and (5) collating, summarising and reporting results. The current review aimed to create an overview of the existing literature by conducting a scoping review about the roles of healthcare leaders in HIT implementation [31]. Because the current review was a scoping review and included no human subjects, no ethical approval was required.

### 2.2. Data Sources and Search Strategy

Database searches were performed by the primary author using CINAHL, Business Source Complete, ProQuest, Scopus and Web of Science. These databases were selected for the review because they contain relevant studies about healthcare leadership. Although the point of scoping reviews is to be as comprehensive as possible [30], PubMed and Google scholar were not used in the current review because other databases provided a voluminous number of records and adding more databases would have probably resulted in duplicates. An information specialist was consulted about the search strategy. Search terms were related to healthcare, HIT and leadership (Table 1). The RefWorks reference system was used to manage citations and remove duplicates. The study selection was independently performed by two reviewers (EL and MH), based on the title and abstract examination and full text examination. Reasons for exclusions were presented. If any disagreement arose, they were solved by consensus between the reviewers. The reference lists of all included studies were manually searched for additional studies. In addition, the Finnish Journal of eHealth and eWelfare was searched manually, since the journal is not indexed in the used databases but is known to contain peer-reviewed studies. The study selection is presented using a PRISMA flow diagram [31] (Figure 1). 

### 2.3. Eligibility Criteria

The original, scientific peer-reviewed articles included in the study were written either in English or Finnish, published between 2000 and 2019, and focused on HIT implementation and contained leadership insight given by various informants (e.g., healthcare managers or leaders, healthcare professionals). Both qualitative and quantitative scientific articles were included to provide greater breadth to the review [30,31]. Those articles that did not have full texts available online, were about the HIT adoption phase, or only contained views of the role of chief information officers (CIOs) were excluded. No grey literature was included. Examples of articles that were not relevant for the review (n = 9) include guidelines for HIT implementation or case studies concerning some specific HIT implementation and did not express leadership insight.

### 2.4. Data Extraction

The data was extracted by author(s), year of publication, country of origin, aim of the study, data and methods, and key findings related to the presented research question [31]. The data was analysed using content analysis, a widely-used method for analysing review material that aims to provide an understanding of the contents of the text and identify the essential themes to answer the research question [32]. 

## 3. Results

Our database and manual search identified a total of 6195 articles after duplicates were removed, and of these 6162 citations were excluded on the basis of the title or abstract. Full-text assessment for eligibility was performed on 33 studies, of which 11 met the inclusion criteria. Also, a manual search of the reference lists of the included studies identified five more studies. Therefore, a total of 16 studies were included in this scoping review (Table 2).

### 3.1. Study Characteristics

The included studies originated from the USA (n = 5), UK (n = 4), Norway (n = 3), Finland (n = 3) and Sweden (n = 1). The informants in the included studies were social and healthcare supervisors and leaders (e.g., middle managers, clinical leaders, project leaders), social and healthcare professionals (e.g., physicians, nurses, midwives, social workers, pharmacists, medical support staff), IT personnel or managers, staff from research institutions, non-governmental organisations or other public sector organisations, persons from innovation and funding agencies, and vendors or external experts. Some studies also used documents and observations as research data. 

Twelve of the included studies were qualitative studies, with most using qualitative content analysis as an analysis method. Three of the studies were mixed or multi-method studies that mostly used quantitative surveys with qualitative interviews. Only one of studies was solely quantitative. 

### 3.2. Healthcare Leaders’ Roles in HIT Implementation

This scoping review identified seven roles for healthcare leaders in HIT implementation. The identified roles were the supporter (n = 11), change manager (n = 10), advocate (n = 7), project manager (n = 7), decision-maker (n = 4), facilitator (n = 3), and champion (n = 3) (Figure 2).

#### 3.2.1. Role of Supporter

According to the included studies, the most common role for healthcare leaders in HIT implementation was that of supporter [33,34,35,38,39,40,41,42,47,48]. Leaders at all levels were responsible for supporting HIT implementation [33,35,39,47]. When leaders did not support implementation, it struggled to succeed [41]. Healthcare leaders often recognised their role as a supporter by themselves [35,39], but occasionally the need for leaders’ support was determined by other stakeholders, such as healthcare professionals [33,43]. It seems that the leaders’ support formed a chain where leaders at a higher level were responsible for supporting their closest subordinates [33,40], for example, clinical leaders were often the ones supporting healthcare professionals and chief executives supported senior managers [33]. The quality of the provided support varied. In some studies, support was described as providing sufficient resources to advance HIT implementation [35,42,46]. These resources were either financial [42] or they were to enable HIT training for the healthcare professionals using HIT [38,48,49]. Support was also understood as motivating healthcare professionals to use HIT and working closely with them [48]. In one study, healthcare leaders even provided technical support for healthcare professionals [46]. In other studies, leaders stated that they felt insecure about their own skills with HIT and required training so they could better guide their colleagues, healthcare professionals and customers [35,39].

#### 3.2.2. Role of Change Manager

Healthcare leaders performed tasks related to change management by informing healthcare professionals about the changes through clear communication [38,42,45] and identifying any resistance [39,41,45]. However, in many cases leading was not simple, for example, physician leaders pointed out that physicians were too autonomous and difficult to lead [48] and resistance to change was common [39,41,45,48]. If health professionals were reluctant to use HIT, managers had to make it a necessary part of their work [37] and they also took an active part in resolving conflicts between stakeholders [34]. Due to the changes, healthcare leaders were also responsible for developing new routines, roles and responsibilities [33,35] and organising the workflow [39].

#### 3.2.3. Role of Advocate

Healthcare leaders also adopted the role of advocate for HIT [33,35,42,44,48]. All leaders, from senior to clinical leaders, agreed that implementing HIT was a high priority project [35,38,44,48]. Healthcare leaders who were firm believers of HIT demonstrated a visible commitment to the implementation process [45]. Hospital management groups frequently pointed out the importance of HIT implementation [44]. This could also lead to problems given that project managers in particular were caught in between IT staff and healthcare professionals, who lacked understanding of each other’s mission [35]. Thus, leaders were also responsible for convincing unwilling new users to view HIT in a more positive way [35,42] and for providing IT staff with a broader understanding of the mission of the healthcare organisation [35]. Healthcare leaders described how HIT would improve patient safety, strengthen an organisation’s core mission, and consolidate its leadership position within the markets [45]. Leaders also reminded health professionals to use HIT and they advocated HIT to their colleagues [48].

#### 3.2.4. Role of Project Manager

Healthcare leaders at all levels were responsible for actively participating in planning the implementation [34,38,47,48], creating implementation teams [35], and occasionally they were responsible for scheduling the implementation [42,44]. Some of the study participants worked as project managers or coordinators, and they felt responsible for the implementation process [38]. Healthcare leaders also felt responsible for the implementation outcomes [48] and were aware of the impacts of the implemented HIT [35].

#### 3.2.5. Role of Decision-Maker

As is natural, healthcare leaders who adopted the role of a manager in HIT implementation [48] were also the ones responsible for making the decision to implement HIT [36,44,45]. The role of a decision-maker was especially common within top level managers [44].

#### 3.2.6. Role of Facilitator

Another common role that healthcare leaders adopted was the role of a facilitator [41]. For example, this role was apparent when healthcare leaders pursued cooperation between the healthcare professional, research centres and vendors [48]. Healthcare leaders also cooperated with health professionals and thus facilitated them to take part in HIT implementation [41]. This helped the group to achieve the goal by providing them with a prerequisite for cooperation [38]. Managers felt responsible for ensuring that nurses and physicians cooperated in the use of HIT [48]. 

#### 3.2.7. Role of Champion

According to the included studies, successful HIT implementation required champions [39]. Occasionally the role of a champion was filled by healthcare leaders [35] and some leaders led by example by being among the first to adopt HIT [45]. In seems that one potential strategy for healthcare organisations is to train leaders to adopt the role of champion [39]. 

## 4. Discussion

This scoping review provided new information about the roles of healthcare leaders in HIT implementation. Altogether, seven roles were identified: supporter, change manager, advocate, project manager, decision-maker, facilitator and champion. Of these roles, the roles of supporter and change manager were the most commonly identified. However, all of the identified roles seem to have an impact on the degree of success of the HIT implementation. 

There are similarities when the roles identified in the present study are compared to leaders’ suggested roles in innovation implementation in a healthcare setting. Birken et al. [23] found that middle managers in innovation implementation were (1) diffusing and synthesising information, (2) mediating between strategy and day-to-day activities, and (3) selling innovation implementation. In HIT implementation, healthcare leaders were found to diffuse information when working as change managers and facilitators [38,41,42,45] and they need to be able to manage tasks that come about as a result of HIT implementation [35,45,48]. Healthcare leaders also act as “sellers” of HIT implementation to their subordinates and colleagues [48] and try to make them see HIT more positively [42]. The importance of leaders’ support was also detected by Hsia et al. [50], who found that top management beliefs support HIT usage in the organisation. Not only is top management support important, but clinical leaders and middle managers also act as supporters [33,35,39,47]. Abbott [6] recognised the role of leaders as champions, and in this role they had the knowledge, skills and understanding of the complexities of HIT, and they were passionate about implementation and better health outcomes. The current review showed that the role of the champion occasionally belongs to subordinates, and leadership should encourage the recruitment of clinical champions and afford them sufficient resources [51]. Results of this kind were also found by Ingebrigtsen et al. [15], who found that identifying and appointing champions was one proactive leadership behaviour associated with successful HIT adoption. They also recognized other actual proactive IT behaviours that are associated with the roles identified in the current review. For example, the behaviour of providing leadership support and establishing training [15] associates with the role of a supporter, in which the healthcare leaders provided support for subordinates and were also responsible for arranging training for them [38,48,49]. Also, communicating clear visions and goals for IT adoption [15] has similarities to the role of an advocate, whereby healthcare leaders described the benefits of HIT to strengthen the organisation’s core missions [45]. Slight similarities were also seen between the behaviour of addressing work process change [15] and the role of a change manager, and between follow-up behaviour [15] and the role of a project manager. Although there are similarities between the role of healthcare leaders in HIT implementation and any other innovation implementation process, HIT implementation has its own unique features, especially its extraordinarily high cost. Healthcare leaders often adopted the role of advocate and prioritised HIT implementation as one of their unit’s major projects [35,38,44,48]. Eagerness to adopt the role as a HIT advocate might be explained by the price of HIT implementation. The price of a CPOE (computerised physician order entry) ranged from USD 3 million to USD 10 million [45], whereas implementing EHR/EMR is much more expensive [52]. Because of the high cost [45] and potential for return-of-investment (ROI) [53], understanding the barriers and facilitators of HIT implementation, such as leadership, is crucial for healthcare organisations [6,12]. 

Healthcare leaders appear to play a major role in a successful HIT implementation [6,12,54]. However, not all leaders are aware of who is responsible for an HIT implementation [39]. Some studies in the present review also proposed that leaders are unable to make HIT implementation a priority because they have to operate other health services [35,48]. Occasionally, some leaders even display a lack of interest in HIT implementation and they do not actively adopt any role in the implementation process [49]. This reluctant leadership might be explained by the lack of confidence among leaders in relation to HIT implementation, and therefore they might prefer to have more support and training for themselves [39]. The included studies reveal a chain of support, where leaders were responsible for supporting their subordinates [33,40]. This kind of behaviour seems to be common, since previous implementation studies have found the same behaviour pattern whereby higher-level leaders supported lower-level leaders [25,55]. 

Providing support for subordinates might be difficult if leaders are not aware of new HIT services themselves. Several studies reveal that healthcare leaders are responsible for arranging training for healthcare professionals [38,43,48], and training is one of the most important factors in a successful HIT implementation [54]. Results from the current study also highlight the importance of sufficient support and training for healthcare leaders themselves. Leaders’ personal information technological competence influences the adoption of HIT by other healthcare professionals [15] and plays a key role in their support [51]. Not only do healthcare leaders seem to have a lack of understanding about HIT implementation, but also the development of policy-level guidance is lagging behind [18]. Thus, the importance of HIT implementation and training should be recognised on a strategic level, and organisational roles and responsibilities should be clearly defined. It has been suggested that in leaders’ training, the following themes should be included: understanding of the existing systems and capabilities of implemented HIT, organisational planning, a committed interdisciplinary team, HIT development with an organisational focus, and HIT implementation support [54]. In addition, leaders should learn about HIT opportunities and actively participate in IT forums and vendor-sponsored conventions, and become more aware of government regulations and policies [50]. Sufficient training and participation might help leaders’ to adapt to complex HIT implementation [56], and help them to adopt the required roles and support the success of HIT implementation by maintaining a positive attitude towards HIT [50]. Some pioneer organisations have already implemented the positions of chief medical informatics officers and chief nursing informatics officers [57], who are specialists in clinical work, management and information technology [20,21]. The recent results of Fenton et al. [57] highlight the requirement for and subsequent development of a doctorate in health informatics (DHI).

## 5. Limitations

This scoping review has a few limitations. First, because of the variable use of terminology in HIT implementation, a comprehensive search strategy was used and it did not use “implementation” as a search term. The comprehensive search strategy resulted in 6195 records, of which only 33 were included for the review whereas the whole point of scoping the field is to be as comprehensive as possible. This may explain the huge number of identified records compared to included ones [30]. Second, only two studies focused explicitly on healthcare leaders’ roles while in other studies the roles were ambiguous. Third, as is the case with most scoping review methodologies, quality assessment was not performed for the included studies [31]. However, because scoping reviews are used to describe the nature of the literature and answer broad research questions, these limitations are believed to be common for scoping reviews.

## 6. Conclusions

Healthcare leaders adopt several different roles in HIT implementation, with supporter, change manager and advocate being the most common roles. Identifying these roles may take us a step closer to successful HIT implementation. However, it seems that healthcare leaders cannot fully achieve these roles, and their understanding of HIT implementation may not be any better than their subordinates’ understanding of it. In addition, healthcare leaders seem to require more support when actively participating in HIT implementation. 

More detailed research is still needed on what actually supports healthcare leaders’ adoption of the roles required for HIT implementation at all leadership levels, and how adopting these roles influences the degree of success of the HIT implementation. In addition, since most of the included studies were qualitative, having more quantitative and mixed methods studies would be beneficial for enriching the research field pertaining to HIT implementation. In particular, validated questionnaires would be highly desirable.

## Figures and Tables

**Figure 1 ijerph-17-02865-f001:**
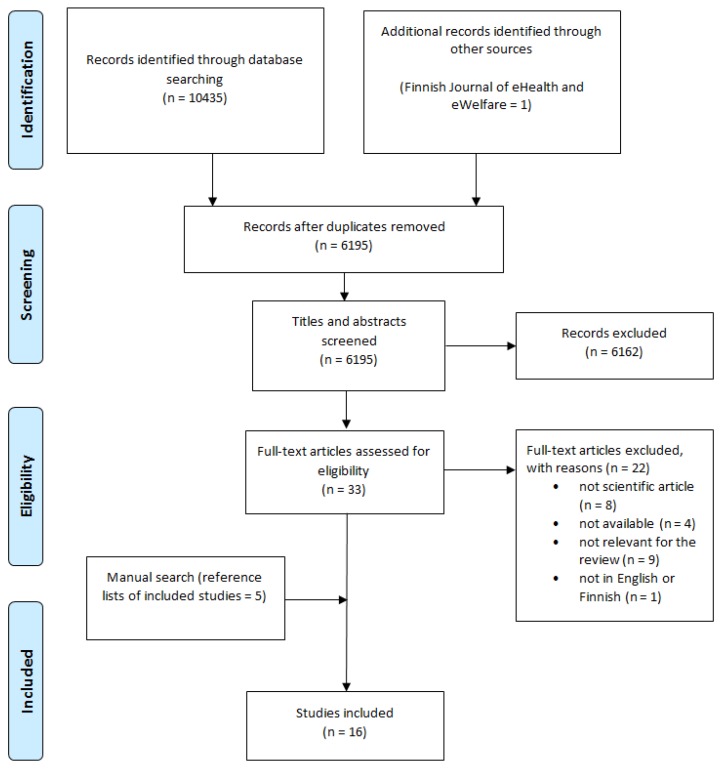
Flow diagram for the scoping review.

**Figure 2 ijerph-17-02865-f002:**
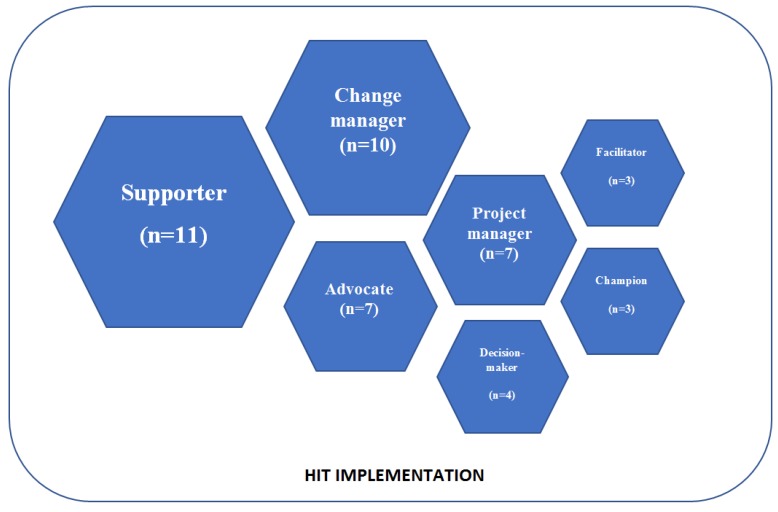
Healthcare leaders’ roles in HIT implementation (n = number of appearances in the included articles).

**Table 1 ijerph-17-02865-t001:** Search strategy.

Keywords	Database
Strategy I	
TI (“health services” OR “health care” OR healthcare OR hospital* OR care) AND TI (“information technology” OR digi* OR “e-health” OR ehealth OR mhealth OR electronic OR telemedicine) AND (leader* OR manage* OR administrat*)	CINAHL (EBSCO) Business Source Complete (EBSCO)
ti (“health services” OR “health care” OR healthcare OR hospital* OR care) AND ti (“information technology” OR digi* OR “e-health” OR ehealth OR mhealth OR electronic OR telemedicine) AND noft (leader* OR manage* OR administrat*)	ProQuest
TITLE (“health services” OR “health care” OR healthcare OR hospital* OR care) AND TITLE (“information technology” OR digi* OR “e-health” OR ehealth OR mhealth OR electronic OR telemedicine) AND TITLE-ABS-KEY (leader* OR manage* OR administrat*)	Scopus
TITLE: (“health services” OR “health care” OR healthcare OR hospital* OR care) AND TITLE: (“information technology” OR digi* OR “e-health” OR ehealth OR mhealth OR electronic OR telemedicine) AND TOPIC: (leader* OR manage* OR administrat*)	Web of Science
Strategy II	
TI (“information technology” OR digi* OR “e-health” OR ehealth OR mhealth OR electronic OR telemedicine) AND TI (leader* OR manage* OR administrat*) AND (“health services” OR “health care” OR healthcare OR hospital* OR care)	CINAHL (EBSCO) Business Source Complete (EBSCO)
Ti (“information technology” OR digi* OR “e-health” OR ehealth OR mhealth OR electronic OR telemedicine) AND ti (leader* OR manage* OR administrat*) AND noft(“health services” OR “health care” OR healthcare OR hospital* OR care)	ProQuest
TITLE (“information technology” OR digi* OR “e-health” OR ehealth OR mhealth OR electronic OR telemedicine) AND TITLE (leader* OR manage* OR administrat*) AND TITLE-ABS-KEY (“health services” OR “health care” OR healthcare OR hospital* OR care)	Scopus
TITLE: (“information technology” OR digi* OR “e-health” OR ehealth OR mhealth OR electronic OR telemedicine) AND TITLE: (leader* OR manage* OR administrat*) AND TOPIC: (“health services” OR “health care” OR healthcare OR hospital* OR care)	Web of Science

TI or ti = Title. noft = Anywhere except full text. TITLE-ABS-KEY = Title, abstract or keywords.

**Table 2 ijerph-17-02865-t002:** Data extraction of the included studies.

The Author(s) (Year) and the Journal	Country of Origin	The Aim of the Study	Data and Methods (Data Collection; Informants; Analysis Method(s))	Key Findings Related to the Research Question
Boddy et al. (2009) [33]	UK	To identify to what extent generic management practices are evident in e-health projects, and to use that knowledge to develop a theoretical model of e-health implementation.	Semi-structured interviews with managers and health professionals (n = 18). Nvivo used for analysis.	Senior manager supported e-health implementation and made it essential to the working practices of senior managers.
Deokar & Sarnikar (2016) [34]	US	To describe how process change issues relate to implementation of large IT projects in healthcare settings.	Data consisted of application reports. Qualitative content analysis.	Management support is critical in EHR implementation. Strong physician and clinical leadership in implementation team were critical in communicating and supporting the goals and vision. Top organisation leaders served on the Leadership Council as well as different project implementation teams. Project implementation team leadership resolved conflicts.
Dugstad et al. (2019) [35]	Norway	To identify the facilitators and barriers for implementation of digital monitoring technology in residential care for persons with dementia and wandering behaviour, and to explore co-creation as an implementation strategy and practice.	Longitudinal case study, interviews (n = 23), strategic documents, participatory observations and process data from workshops (n = 7), observations of local training sessions and numerous meetings. Content analysis.	Healthcare leaders are responsible for developing new routines, roles and responsibilities. In addition, allocating sufficient time and resources across roles and professions for workshops and other implementation strategies proved to be a facilitator. The leaders’ priority was operating the service. Project managers provided technical support and filled the role of implementation champions.
Hall et al. (2017) [36]	UK	To explore facilitators and barriers to the implementation of monitoring technologies in care homes.	Semi-structured interviews of staff, relatives and residents (n = 36), observation, resident care record view. Framework analysis.	Senior management made decisions to implement HIT.
King et al. (2012) [37]	UK	To explore the way in which structural, professional and geographical boundaries have affected e-health implementation in health and social care.	Interviews of health and social care professionals (n = 30) and telephone interviews (n = 11). Framework analysis.	Managers made the decision to make SSA a necessary part of the referral process, when healthcare professionals were reluctant to use it.
Kujala et al. (2018) [38]	Finland	To identify good implementation practices and understand their use.	Survey-based data from supervisors and leaders (n = 478). Interviews with four project managers or coordinators. Descriptive statistics and content analysis.	The identified good practices were communicating clear leadership support, informing about the service implementation and its benefits, and user participation in planning.
Kujala et al. (2019) [39]	Finland	To evaluate clinical leaders’ eHealth competencies and training needs in two public healthcare organisations in Finland.	Online survey of clinical leaders (n = 98). Descriptive statistics and content analysis.	Clinical leader had critical role in supporting healthcare professionals and avoiding resistance to change.
Kujala et al. (2019) [40]	Finland	To examine whether frontline leaders’ positive expectations of a patient portal and perceptions of its implementation were associated with their support of the portal. To explore whether leaders’ positive perceptions influenced the same unit’s health professional support for the portal.	Online survey of 2067 health professionals and 401 frontline leaders. Several descriptive statistics and reliability analyses.	Healthcare leaders participated in the planning of patient portal service. Leaders’ clear vision of the patient portal was moderately associated with their support for the portal.
Mason et al. (2017) [41]	UK	To explore rural primary care physicians and physician assistants’ experiences regarding overcoming barriers to implementing electronic health records.	Interviews with physicians and physician assistants (n = 21). Phenomenological research analysis and narrative segments.	EHR implementation struggles when managers do not support it. The collaboration between healthcare leaders and providers might enhance the degree of operational, technological, clinical and financial success.
McAlearney et al. (2014) [42]	USA	To comprehensively study and synthesise best practices for managing ambulatory EHR system implementation in healthcare organisations, highlighting applicable management theories and successful strategies.	Interviews (n = 45) with key informants and six focus groups comprised of 37 physicians. Both deductive and inductive analysis methods.	Five factors that appear to facilitate successful management of HIT implementation were characterised: (1) commitment; (2) convincing/converting; (3) communication; (4) coordination; and (5) change management.
Nilsen et al. (2016) [43]	Norway	To identify and describe forms of resistance that emerged in five municipalities during a technology implementation project as part of the care for older people.	Data from interviews with focus groups (21 individuals, both healthcare providers and technology developers) and participatory observation (about 50 individuals, including five researchers). Kvale’s description of the bricolage approach and research triangulation.	Project managers and healthcare professionals experienced a lack of interest and support from middle managers, unit leaders and ward nurses. The need for training was recognised by project leaders and other participants, but responsible leaders did not arrange this.
Øvretveit et al. (2007) [44]	Sweden	To describe an implementation of one information technology system in one hospital, the perceived impact, the factors thought to help and hinder implementation, and the success of the system, comparing this with theories of effective IT implementation.	Qualitative case study using semi-structured interviews (n = 30) and documentation. Participants: senior clinicians, managers, project team members, doctors and nurses. Thematic analysis.	Top leadership was responsible for making a timetable and managing project tightly. Senior leaders set their date for implementation. Senior managers and heads of the clinics felt that HIT implementation was their highest priority. Hospital management group pointed out the importance of the project.
Poon et al. (2004) [45]	USA	N/A	Semi-structured interviews of senior managers (n = 52). Grounded-theory approach.	Overcoming resistance requires strong leadership. Healthcare leaders had to be firm believers of CPOE and they need to be able to manage changes that come with implementation. Some managers were among the first to adopt CPOE.
Stevenson et al. (2018) [46]	USA	To provide guidance and support for the implementation and spread of SCAN-ECHO.	Mixed-methods approach involving two quantitative surveys and qualitative interviews (n = 52). A consensual qualitative analysis.	Leaders provided technical support and gave resources for training session.
Szydlowski & Smith (2009) [47]	USA	To examine the trends of healthcare leadership and management with regard to implementation and management of IT in a hospital setting.	Interviews (n = 12) with CIOs and nurse managers. Comparative analysis.	Nurse managers thought that chief executive officer’s leadership and support of the HIT process increase the probability of efficient and effective HIT implementation.
Varsi et al. (2015) [48]	Norway	To examine the perceptions of nurse and physician managers regarding facilitators, barriers, management role, responsibility, and action taken in the implementation of an eHealth intervention called Choice into clinical practice.	A qualitative study with descriptive design based on individual interviews with nurse (n = 6) and physician managers (n = 3). Content analysis.	Managers supported the implementation, established collaboration between different actors and took the initiative to arrange training sessions. Managers also had Choice regularly on the agenda for their management meetings and managers spent time reminding nurses to use Choice and recommended it to colleagues. Managers felt it was their responsibility to ensure the implementation.

IT = information technology; EHR = electronic health record; SCAN-ECHO = Specialty Care Assess Network-Extension for Community Healthcare Outcomes; CIO = chief information officer; Choice = interactive tailored eHealth intervention for patient assessment; CPOE = computerised physician order entry.

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
