# Peer review of "Identifying the Roles of Healthcare Leaders in HIT Implementation: A Scoping Review of the Quantitative and Qualitative Evidence"

_ijerph, 2020, doi:10.3390/ijerph17082865_

Round 1
Reviewer 1 Report
The introduction is short but seemingly to the point.
Data search strategy: the database chosen seem appropriate but the authors might want to comment on why Google scholar and pubmed was left out? In a scoping review where the quality of evidence is secondary to finding the width of the problem under scrutiny this could be seen as a valid strategy.
Search strings and search strategies seems legitimate but looking at the flowchart where the drop from 6195 to an eligible number of 33 probably a bit loose. It is however not uncommon in this kind of search but a comment on this in the discussion on methods will be welcome. It could also be interesting to know something about the nine articles not relevant for the review that was read in full text but excluded.
The choice of a scoping review seems appropriate since the research question concerns the identification of certain characteristics/concepts in sources of evidence, and in the mapping, reporting or discussion of these characteristics/concepts.
The result is mostly presented in a clear and relevant way but there are some problems with the description (or possibly the naming) of the roles of healthcare leaders. The three roles as managers become a bit confusing and especially 3.2.5 role of manager is insufficiently explained and described and lacks orthogonality towards many of the other roles. Since most of the roles are to some extent managerial roles maybe this role should be called decision-maker or something similar (if that is the main characteristics of this role) It also becomes apparent in the results section that the authors have not given any definition or explicit description of their bearing concept of healthcare leaders.
Anyway please provide clarifications and definitions to improve the results section.
Author Response
Dear reviewer,
On behalf of myself and my fellow-authors, I would like to thank you for your interesting and constructive points of view regarding to our manuscript of ‘Identifying the Roles of Healthcare Leadership in HIT implementation: A Scoping Review of the Quantitative and Qualitative Evidence’. We agreed with all the comments and used them to improve the manuscript. In below we explained how we took the comments into account. None of the reviewers suggested the manuscript to undergo extensive English editing, but I wanted to let you know, that the language has been checked prior submission by using a native English speaker.
- We applied the mention why we did not use PubMed and Google Scholar to rows 80-84. We also added a comment about the drop from 6195 studies to an eligible number of 33 studies to limitations (rows 289-292). We also explained what were the nine articles that were defined as "not relevant for the review" (rows 105-107).
- When analyzing the results, we were also had two choices for the role of manager, and the other was the role of decision-maker. For the current manuscript we decided to rename this role as the role of decision-maker. This change has been notified in whole text (incl. figures).
- We have also explained the concept of healthcare leadership in the manuscript (rows 48-49).
Best regards,
Elina Laukka
Reviewer 2 Report
The authors raise important issues in the article.
The article is well structured.
Research questions were put and the goal was formulated.
In the introduction, it would be worth emphasizing the added value of the research contained in the article.
Author Response
Dear Reviewer,
On behalf of myself and my fellow-authors, I would like to thank you for your interesting and constructive points of view regarding to our manuscript of ‘Identifying the Roles of Healthcare Leadership in HIT implementation: A Scoping Review of the Quantitative and Qualitative Evidence’. We agreed with all the comments and used them to improve the manuscript. In below we explained how we took the comments into account. None of the reviewers suggested the manuscript to undergo extensive English editing, but I wanted to let you know, that the language has been checked prior submission by using a native English speaker.
We added the value of the research to 'Introduction' (rows 48-49).
Best regards,
Elina Laukka
Reviewer 3 Report
This article is a step in the right direction on a complex, timely and important issue. Implement Health Information Technology leadership at all levels is indispensable to improve health systems performance. The authors analyze well the key issues needed to define the research question in the introduction. The question is somewhat generic, but as they explain well in the design section and in the limitations is part of the nature of this type of research design when using a scoping review. In the “Results” section, I think it would be helpful for the reader to see the relation of the roles identified by the authors with the similar ones that have been identified before and described in the introduction of the paper. This will help the reader understand better the “leadership” actions and the potential difference with other related management roles that could be involve in the “implementation” strategies or processes that sometimes are more concrete and operational. That could also be useful to get the reader thinking about the competencies needed to be developed and possible training strategies to strengthen some of the roles described in the paper later in the “Discussion” section.
This paper provides the authors and other researchers with ideas to continue developing related research to advance this topic.
Author Response
Dear Reviewer,
On behalf of myself and my fellow-authors, I would like to thank you for your interesting and constructive points of view regarding to our manuscript of ‘Identifying the Roles of Healthcare Leadership in HIT implementation: A Scoping Review of the Quantitative and Qualitative Evidence’. We agreed with all the comments and used them to improve the manuscript. In below we explained how we took the comments into account. None of the reviewers suggested the manuscript to undergo extensive English editing, but I wanted to let you know, that the language has been checked prior submission by using a native English speaker.
Your point about comparing to our results with Ingebrigtsen et al. was interesting and we did not notice this earlier. However, we saw the suitable place for the comparison in 'Discussion' (rows 231-242) instead of 'Results'. In 'Discussion' this comparison also invites readers to think about competencies that need to be developed and training strategies.
Best regards,
Elina Laukka